# WAPITI: A Watermark for Finetuned Open-Source LLMs

## Abstract

Watermarking of large language model (LLM) generations embeds impercepti­ble statistical patterns within text, enabling algorithmic detection. It provides a promising defense for ensuring traceability, accountability, and integrity of open-source models. However, current watermarking approaches face two key limita­tions: incompatibility with fine-tuned models and intense training cost. In this work, we propose WAPITI, a watermark framework tailored for fine-tuned mod­els. Our contributions are threefold: (1) We introduce a train-efficient watermark­ing that eliminates the need for large domain-specific datasets and requires sub­stantially less training. (2) We enable seamless integration of our framework with existing watermarking techniques, making it broadly compatible with diverse wa­termarking schemes. (3) We provide an in-depth empirical analysis of the mech­anism underlying watermark transfer, offering insights into how parameter-level operations influence both watermark strength and model capabilities. Extensive experiments across architectures and watermarking strategies demonstrate that WAPITI effectively injects watermarks into fine-tuned models while preserving their adapted capabilities and robustness.

## 1 Introduction

Large language models (LLMs; Touvron et al., 2023; OpenAI et al., 2024) have been integrated into many workflows and now play an increasingly important role in daily life. This rapid adoption also raises concerns: it is often difficult to distinguish LLM-generated text from human-written content, which may lead to misinformation or misuse. To address this, **watermarking** has been proposed as a promising solution. Watermarking embeds hidden signals in model outputs that can later be detected, enabling reliable identification of AI-generated text. This not only allows users to separate AI content from human content for verification but also makes it possible to trace text back to the source model, providing a technical foundation for regulatory oversight of language models.

Most prior work on watermarking has focused on closed-source models (Kirchenbauer et al., 2023; Aaronson, 2023; Kuditipudi et al., 2024), which are black boxes to users. In this setting, the threat model assumes adversaries try to remove the watermark without access to the model's internal struc­ture; they can only modify generation hyperparameters or apply text post-processing. Methods de­signed for closed-source models usually insert watermark signals by adding extra components into the model (Liu et al., 2024).

With the rise of powerful open-source models (Grattafiori et al., 2024; Qwen et al., 2025) and their many finetuned variants, oversight of open-source models is equally important. However, the key challenge is that the threat model here is fundamentally different: adversaries can access the full model parameters and structure. This makes watermarking methods developed for closed-source models difficult to adapt to open-source settings.

For open-source watermarking, Gu et al. (2024) has proposed using model distillation to embed watermarks into models. However, this approach faces a serious limitation: it is incompatible with fine-tuning. As shown in Gu et al. (2024), when a distilled model is finetuned on non-watermarked data, its watermark quickly disappears. We further extend this setting and provide comprehensive evidence of the incompatibility between watermark distillation and the fine-tuning process. In ad­dition, distillation itself is resource-intensive. For example, current watermark distillation requires

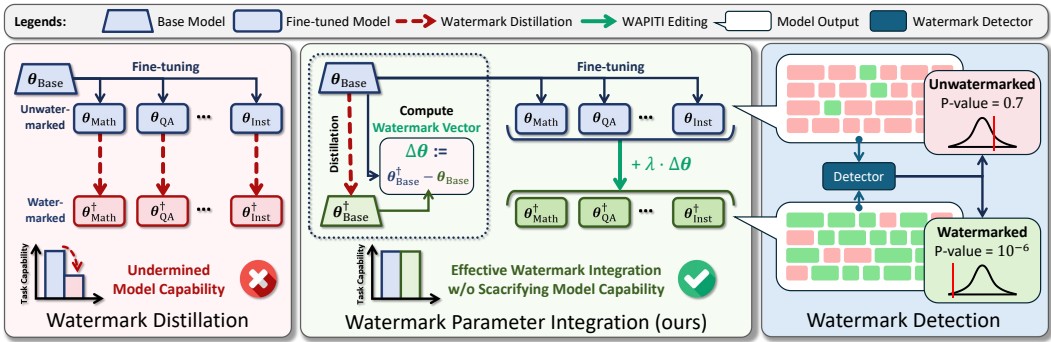

Figure 1: Watermark distillation (left) impairs models' finetuned capabilities. WAPITI (middle) uses watermark parameters to transfer watermarking from the base model to finetuned models. This method can preserve the finetuned model's capabilities while enabling it to generate watermarked texts, where the green tokens indicate the watermarked tokens (right).

nearly 20M tokens (batch size 16, 256 tokens per step). Considering the countless finetuned model variants, this overhead becomes prohibitive.

To address these limitations and make watermarking more practical for open-source models, we propose a training-efficient strategy called **WAPITI** *(WAtermark Parameter InTegratIon)*. Instead of repeatedly distilling each finetuned model, WAPITI transfers watermark information from a distilled base model to finetuned models using parameter integration (see Figure E.2). This approach significantly reduces the cost: watermark distillation is performed only once on the base model, and the resulting watermark parameters can then be reused. (See Table 1 for overall comparison)

Our theoretical analysis and empirical experiments show that integrated finetuned models retain their task-specific performance while exhibiting clear watermark signals. Moreover, by adjusting the coefficient of injected watermark parameters, we can further reduce the training cost of the initial distillation, lowering the overall computational burden.

Our main contributions are as follows:

- **Problem.** We systematically identify the vulnerability of watermark distillation: its incompatibility with fine-tuning, a key obstacle for watermarking open-source models.

- **Method.** To the best of our knowledge, we propose the first transfer-based watermarking method for finetuned models (WAPITI). Our design is based on the observation that watermarking causes an aligned distribution shift before and after distillation.

- **Analysis.** We provide a theoretical analysis of WAPITI's utility and examine the relationship between watermark parameters and finetuned vectors to explain the mechanism of watermark transfer.

- **Evaluation.** We evaluate WAPITI on Llama-2-7B, Llama-3-8B, and Qwen-2.5-3B. We select medical QA and legal summarization as fine-tuning tasks. WAPITI achieves high detectability, with a true positive rate (TPR) of 0.98 at a false positive rate (FPR) of 0.05, while successfully retaining finetuned performance.

## 2 PRELIMINARIES OF WATERMARKING

Large Language Models (LLMs) are typically neural networks based on the transformer architecture. Formally, we denote an LLM as $f_{\boldsymbol{\theta}} : \mathcal{V}^* \rightarrow \Delta(\mathcal{V})$, which maps a prefix string $\boldsymbol{x} \in \mathcal{V}^*$ to a probability distribution over the vocabulary $\Delta(\mathcal{V})$ for predicting the next token. The conditional distribution of the next token is written as $f_{\boldsymbol{\theta}}(\cdot \mid \boldsymbol{x})$. The generation process involves two main steps: *logit generation* followed by *token sampling* (Vaswani et al., 2017).

Watermarking modifies the generation process so that hidden, traceable information is embedded into the output text. This is usually done during decoding, either at the logit stage or the sampling stage, in a way that guides the output distribution toward patterns recognizable by a detector. For

| Method | Closed-source | Open-sourced | | Open-sourced Application | |
|--------|:---:|:---:|:---:|:---:|:---:|
| | LLMs | Base LLMs | Fine-tuned LLMs | Efficiency | Vulnerability |
| Decoding-based | ✓ | ✗ | ✗ | N/A | Architecture Modification |
| Distillation-based | N/A | ✓ | ✗ | $\mathcal{C}_{FT}$ | Finetune Incompatibility |
| **WAPITI** | N/A | N/A | ✓ | $\mathcal{C}_{FT}/N$ | N/A |

Table 1: A taxonomy of LLM watermarking. "N/A" indicates that the method is not designed for the corresponding setting. And $\mathcal{C}_{FT}$ indicates the computation cost of watermark distillation. $N$ denotes the number of models of the same type, highlighting that WAPITI requires only a single watermark distillation to apply across all models of that type.

example, KGW (Kirchenbauer et al., 2023) increases the probability of certain tokens during generation. A detector can then identify AI-generated text based on the frequency of token occurrence.

Formally, a watermarking algorithm $\mathcal{W}$ uses a secret key $\phi$ to modify the original output distribution $f_{\boldsymbol{\theta}}(\cdot \mid \boldsymbol{x})$ into a watermarked version. A detector $\mathcal{D}$, given the same key $\phi$, attempts to recover the embedded signal. For a given text $x$ and key $\phi$, the detector computes a p-value under the null hypothesis that $x$ is not watermarked. If the p-value is below a predefined threshold, the text is classified as model-generated. Further details are provided in Appendix A.

The key evaluation metrics for watermarking are: (i) **Detectability:** The watermark should allow reliable detection of all model-generated outputs. (ii) **Utility:** The watermark should not significantly degrade the model's original performance. (iii) **Security:** The watermark should be hard to remove without heavily altering the output. For open-source models, removal should not be possible without impairing their overall capability.

**Logit-based watermarking (KGW).** This method directly modifies output logits (Kirchenbauer et al., 2023). At each step, the vocabulary is split into *green* and *red* lists based on the previous $k$ tokens. For $k = 0$ (Zhao et al., 2023b), the split is fixed; for $k \geq 1$, it depends on context. A fraction $\gamma$ of tokens are marked green, and their logits are increased by $\delta$, making them more likely to be sampled. Detection tests whether the observed proportion of green tokens exceeds $\gamma$, yielding a p-value.

**Sampling-based (AAR).** This method (Aaronson, 2023) applies Gumbel-Max sampling. For token $x_i$, the previous $k$ tokens and key $\phi$ generate a pseudorandom score vector $\boldsymbol{r}_i \in [0, 1]^{|\mathcal{V}|}$. Given next-token distribution $\boldsymbol{p}_i \in \Delta(\mathcal{V})$, AAR samples

$$x_i = \arg\max_{j \in \mathcal{V}} \big( \log p_{i,j} - \log(-\log r_{i,j}) \big),$$

introducing controlled randomness. This yields higher cumulative score sums in watermarked text, so detection uses the score sum to compute a p-value under the null hypothesis.

## 3 LIMITATIONS OF WATERMARK DISTILLATION

A key characteristic of open-source models is their **flexibility**: base models can be fine-tuned to gain new capabilities. Therefore, for watermarking methods designed for open-source models, compatibility with fine-tuning is critical. In this section, we investigate whether watermark distillation (Gu et al., 2024) can embed watermarks into fine-tuned models. While Gu et al. (2024) briefly noted that fine-tuning could be considered an attack against watermarks, we extend their setting and systematically evaluate whether watermark distillation remains compatible with fine-tuning. In simple terms, the goal is to obtain a model that preserves watermark detectability while also retaining strong fine-tuned performance.

**Setup.** We evaluate three strategies for integrating watermarking with fine-tuning: (i) fine-tuning a watermark-distilled model on the domain dataset, (ii) applying watermark distillation to a model that has already been fine-tuned, and (iii) paraphrasing the domain dataset with a watermarked model and then fine-tuning on the resulting watermarked corpus. These settings cover the main possible orders of applying watermarking and fine-tuning. The full experimental details are provided in Appendix C.

(i) finetune a watermark-distilled model on the domain dataset. (ii) Apply watermark distillation to an already fine-tuned model. (iii) Use a watermarked model to paraphrase the domain dataset and finetune on the resulting watermarked corpus.

Our experiments use the Llama-3.1-8B-Instruct model as the backbone, ensuring alignment with state-of-the-art instruction-tuned models. We adopt the KGW watermarking scheme (Kirchenbauer et al., 2023) and select legal summarization (Shen et al., 2022) as the fine-tuning task, as it is a high-entropy generative domain well-suited for testing watermark detectability.

**Analysis.** Figure 2 shows the results for the three strategies, along with WAPITI.

*Approach 1 (watermark-then-finetune).* As reported in Gu et al. (2024), fine-tuning a watermarked model on non-watermarked content quickly degrades watermark strength. We observe the same effect: the model's watermark detectability drops to a p-value of 0.45, close to random (0.5), effectively erasing the watermark.

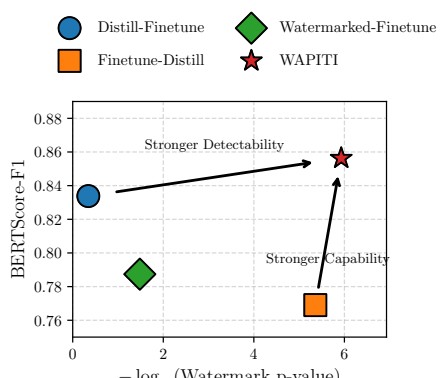

Figure 2: Watermark distillation cannot ensure both strong detectability and fine-tuning performance.

*Approach 2 (finetune-then-watermark).* Applying watermark distillation after fine-tuning causes the model to forget its domain-specific knowledge. Since watermark distillation itself is essentially a form of fine-tuning, this leads to catastrophic forgetting (Luo et al., 2025). The resulting model performs similarly to the base model, losing most of the gains from domain fine-tuning.

*Approach 3 (joint watermark–finetune).* Here we paraphrase the domain dataset with a watermarked model and finetune on the watermarked data. However, this approach also fails: the resulting model achieves neither strong watermark detectability nor strong fine-tuned performance. We attribute this to two reasons: (1) the paraphrased dataset is of lower quality than the original dataset, and (2) the domain dataset is relatively small, less than 10% of the size of our watermark distillation dataset, making it insufficient to sustain both watermarking and fine-tuning.

Overall, our findings show that watermark distillation fails to embed watermarks into fine-tuned models, as it can't simultaneously preserve watermark detectability and domain-specific knowledge.

## 4 METHOD: WATEMRRARK PARAMETER

In this section, we focus on deriving the watermarked parameters of fine-tuned models. As mentioned in §2, watermarks only perturb the next-token generation $x_t$ according to previous $k$ tokens $x_{t-k}, \cdots, x_{t-1}$ and watermark key $\phi$, so that watermark perturbation in next-token probability $f_{\boldsymbol{\theta}}(\boldsymbol{x})^1$ remains the same across different models, where $\boldsymbol{x}$ is the input prompt. We denote the watermark perturbation as $\delta \cdot g(\boldsymbol{x})$, where $\delta$ represents the intensity of the shift, analogous to the watermark shift $\delta$ in KGW and $g(\boldsymbol{x})$ is analogous to the mask of green list in KGW watermarking that indicates which part of vocabulary will be applied watermark shift. Let $\boldsymbol{\theta}_{\text{Base}}, \boldsymbol{\theta}_{\text{Base}}^{\dagger}$ represent parameters of the base model and the watermark-distilled base model, respectively. So we have:

$$f_{\boldsymbol{\theta}_{\text{Base}}^{\dagger}}(\boldsymbol{x}) = f_{\boldsymbol{\theta}_{\text{Base}}}(\boldsymbol{x}) + \delta_{\text{Base}} \cdot g(\boldsymbol{x}). \tag{1}$$

Similarly, we use $\boldsymbol{\theta}_{\text{FT}}$ and $\boldsymbol{\theta}_{\text{FT}}^{\dagger}$ to represent the parameters of the fine-tuned (FT) models, as well as their watermark-distilled counterparts, respectively. Our ultimate goal is, given an unwatermarked $\boldsymbol{\theta}$, to find the parameter $\boldsymbol{\theta}_{\text{FT}}^{\dagger}$ such that:

$$f_{\boldsymbol{\theta}_{\text{FT}}^{\dagger}}(\boldsymbol{x}) = f_{\boldsymbol{\theta}_{\text{FT}}}(\boldsymbol{x}) + \delta_{\text{FT}} \cdot g(\boldsymbol{x}), \tag{2}$$

where $\delta_{\text{FT}}$ is a hyperparameter that controls the watermark detectability.

---

[1] For brevity, we identify the next-token probability predictor $f_{\boldsymbol{\theta}}(\cdot \mid \boldsymbol{x}) : \mathcal{V} \to \mathbb{R}$ as a vector $f_{\boldsymbol{\theta}}(\boldsymbol{x}) \in \Delta(\mathcal{V})$.

Then by using the assumption from previous research (Ilharco et al., 2023; Jiao et al., 2024; Yang et al., 2024), we find that the gradient difference between the fine-tuned and base models, when multiplied by the watermarked parameter difference of bthe ase model, is approximately zero. The detailed derivation of the assumption can be found in Append B:

$$(\nabla_{\boldsymbol{\theta}} f_{\boldsymbol{\theta}_{\text{FT}}}(\boldsymbol{x}) - \nabla_{\boldsymbol{\theta}} f_{\boldsymbol{\theta}_{\text{Base}}}(\boldsymbol{x})) \Delta\boldsymbol{\theta}_{\text{Base}} \approx 0. \tag{3}$$

By rearranging Eq. (18), we conclude that the gradients of the fine-tuned and base models are approximately equal when applied to the watermarked parameter difference:

$$\nabla_{\boldsymbol{\theta}} f_{\boldsymbol{\theta}_{\text{FT}}}(\boldsymbol{x}) \Delta\boldsymbol{\theta}_{\text{Base}} \approx \nabla_{\boldsymbol{\theta}} f_{\boldsymbol{\theta}_{\text{Base}}}(\boldsymbol{x}) \Delta\boldsymbol{\theta}_{\text{Base}}. \tag{4}$$

In this way, we obtain the relationship between the gradient of the fine-tuned model and base models. And we now proceed to derive our target $f_{\boldsymbol{\theta}_{\text{FT}}}(\boldsymbol{x})$. First, by substituting $g(\boldsymbol{x})$ from Eq. (1) into Eq. (2) and use Eq. (19:

$$f_{\boldsymbol{\theta}_{\text{FT}}^{\dagger}}(\boldsymbol{x}) = f_{\boldsymbol{\theta}_{\text{FT}}}(\boldsymbol{x}) + \left( \frac{\delta_{\text{FT}}}{\delta_{\text{Base}}} \langle \nabla_{\boldsymbol{\theta}} f_{\boldsymbol{\theta}_{\text{Base}}}(\boldsymbol{x}), \Delta\boldsymbol{\theta}_{\text{Base}} \rangle + O(\|\Delta\boldsymbol{\theta}_{\text{Base}}\|^2) \right). \tag{5}$$

We define $\lambda_{\text{FT}} = \frac{\delta_{\text{FT}}}{\delta_{\text{Base}}}$, where $\delta_{\text{FT}}$ is a hyperparameter, making $\lambda_{\text{FT}}$ a tunable factor. Next, we substitute the gradient of base model in Eq. (5) with the gradient of fine-tuned model using Eq. (19):

$$f_{\boldsymbol{\theta}_{\text{FT}}^{\dagger}}(\boldsymbol{x}) \approx f_{\boldsymbol{\theta}_{\text{FT}}}(\boldsymbol{x}) + \langle \nabla_{\boldsymbol{\theta}} f_{\boldsymbol{\theta}_{\text{FT}}}(\boldsymbol{x}), \lambda_{\text{FT}} \cdot \Delta\boldsymbol{\theta}_{\text{Base}} \rangle + O\left(\|\Delta\boldsymbol{\theta}_{\text{Base}}\|^2\right), \tag{6}$$

$$\approx f_{\boldsymbol{\theta}_{\text{FT}} + \lambda_{\text{FT}} \cdot \Delta\boldsymbol{\theta}_{\text{Base}}}(\boldsymbol{x}). \tag{7}$$

We treat Eq. (6) as a Taylor expansion of the next-token probability of the model with respect to its parameters. Based on Eq. (7), we can select:

$$\boldsymbol{\theta}_{\text{FT}}^{\dagger} := \boldsymbol{\theta}_{\text{FT}} + \lambda_{\text{FT}} \cdot \Delta\boldsymbol{\theta}_{\text{Base}}. \tag{8}$$

According to derivation, we propose **WA**termark **P**arameter **InT**egrat**I**on (WAPITI), which integrates watermark-related parameters of base model to fine-tuned models. The algorithm is shown in Alg. 1. WAPITI is compatible with various watermarking strategies: after distilling a base model with the desired watermark (Step 1), the watermark can be seamlessly transferred to fine-tuned models without additional costs (Step 3). This approach provides an efficient and effective solution for regulating open-source models.

---

**Algorithm 1** WAPITI

**Input:** base model parameter $\boldsymbol{\theta}_{\text{Base}}$, fine-tuned model parameter $\boldsymbol{\theta}_{\text{FT}}$, watermark coefficient $\lambda_{\text{FT}}$
**Output:** watermarked fine-tuned model parameter $\boldsymbol{\theta}_{\text{FT}}^{\dagger}$

1: $\boldsymbol{\theta}_{\text{Base}}^{\dagger} \leftarrow \text{WatermarkDistillation}(\boldsymbol{\theta}_{\text{Base}})$
2: $\Delta\boldsymbol{\theta}_{\text{Base}} \leftarrow \boldsymbol{\theta}_{\text{Base}}^{\dagger} - \boldsymbol{\theta}_{\text{Base}}$
3: $\boldsymbol{\theta}_{\text{FT}}^{\dagger} \leftarrow \boldsymbol{\theta}_{\text{FT}} + \lambda_{\text{FT}} \cdot \Delta\boldsymbol{\theta}_{\text{Base}}$

---

## 5 EXPERIMENTS

### 5.1 EXPERIMENT SETUP

We design experiments to evaluate the performance of WAPITI from two perspectives: *watermark strength* and *fine-tuning ability*, tested across multiple models and watermarking strategies.

**Watermarks and hyperparameters.** Since our framework is compatible with different watermarking schemes, we select two representative watermarking method spanning logit-based watermark KGW, and sampling-based watermark AAR, each with standard hyperparameter settings. To ensure fair comparison, we adopt the configurations used by Gu et al. (2024). Specifically, for KGW we set $k = \{0, 1\}$, $\gamma = 0.25$, and $\delta = \{1, 2\}$ and select AAR's hyperparameter $k = \{2, 3, 4\}$ The watermark coefficient, $\lambda_{\text{FT}}$, is varied in the range $[0, 4]$.

**Datasets and models.** To evaluate the generality of WAPITI, we experiment on three widely used open-source LLMs: Llama-2-7B, Llama-3.1-8B, and Qwen2.5-3B. These models differ in size

and architecture, and their popularity ensures the practical relevance of our results. For watermark distillation, we use the RealNewsLike subset of the C4 dataset (Raffel et al., 2020). To avoid data leakage and ensure valid detectability results, we apply deduplication to remove overlapping samples before distillation.

**Training parameters.** We adopt standard fine-tuning settings with AdamW optimizer, linear warmup, and cosine decay scheduling. Each model is trained for 5k steps with a moderate batch size and bf16 mixed precision. Full hyperparameter details, including learning rates, prompt construction, and hardware configurations, are provided in Appendix E.

## 5.2 EVALUATION METRICS

We assess WAPITI using three criteria: *watermark detectability*, *generation quality*, and *downstream fine-tuned performance*. These capture both watermark strength and the preservation of model abilities across instruction-following, question answering, and summarization tasks.

**Watermark detectability.** We measure detectability using the median p-value and the true positive rate (TPR) at fixed false positive rate (FPR) thresholds. The p-value is computed using the z-score method; a lower p-value indicates stronger detectability. TPR is computed using equal-sized sets of human-written and watermarked model outputs, truncated to the same length for consistency. We report TPR at FPR values of 0.05 and 0.1 to reflect watermark applicability under realistic detection settings.

**Generation quality.** We assess generation quality using two metrics: perplexity and seq-rep-3 (trigram repetition). Perplexity is computed with Llama-3.1-8B-Instruct (Grattafiori et al., 2024) to ensure consistency across models. Seq-rep-3 quantifies repetition by measuring the proportion of repeated trigrams in the generated outputs (Welleck et al., 2019). For evaluation, we use the OpenWebText dataset (Gokaslan & Cohen, 2019), which differs from the watermark distillation dataset, ensuring that the results reflect the generalizability of WAPITI.

**Fine-tuning abilities.** To verify that WAPITI preserves task performance, we evaluate on: i) *Question Answering:* MedicalQA (Divi, 2023), with 6.3k domain-specific medical questions. ii) *Summarization:* Multi-LexSum (Shen et al., 2022), containing 9.2k legal case summaries. We use *Bertscore* (Zhang et al., 2019) as a metric since we are testing on open-ended generation tasks.

## 5.3 RESULTS

| Scheme | Model | Watermark Detectibility | | | | Generation Quality | | | | Training Cost | |
| | | p-value(↓) | | TPR@0.05(↑) | | Perplexity(↓) | | seq-rep-3(↓) | | GPU Hours | |
| | | WD | WAPITI | WD | WAPITI | WD | WAPITI | WD | WAPITI | WD | WAPITI |
|---|---|---|---|---|---|---|---|---|---|---|---|
| **KGW** | Llama-2 | $3.60 \times 10^{-06}$ | $3.80 \times 10^{-15}$ | 1.00 | 1.00 | 6.17 | 10.37 | 0.05 | 0.05 | 32.00 | 1.60 |
| | Llama-3.1 | $6.05 \times 10^{-08}$ | $2.36 \times 10^{-18}$ | 0.81 | 1.00 | 8.58 | 15.34 | 0.04 | 0.04 | 32.00 | 1.60 |
| | Qwen2.5 | $7.19 \times 10^{-06}$ | $4.67 \times 10^{-13}$ | 0.98 | 1.00 | 10.20 | 14.15 | 0.03 | 0.04 | 6.00 | 0.30 |
| **AAR** | Llama-2 | $3.62 \times 10^{-12}$ | $4.13 \times 10^{-14}$ | 0.91 | 0.89 | 17.32 | 21.21 | 0.04 | 0.05 | 32.00 | 1.60 |
| | Llama-3.1 | $9.13 \times 10^{-12}$ | $6.89 \times 10^{-16}$ | 0.94 | 0.88 | 12.43 | 15.31 | 0.03 | 0.04 | 32.00 | 1.60 |
| | Qwen2.5 | $3.21 \times 10^{-13}$ | $1.89 \times 10^{-14}$ | 0.90 | 0.90 | 17.68 | 16.68 | 0.04 | 0.03 | 6.00 | 0.30 |

Table 2: Main results on watermark detectability, generation quality, and training cost (GPU hours) for WAPITI compared with traditional Watermark Distillation (WD). WAPITI achieves stronger detectability with significantly lower computation cost.

**Performance Comparison** We first compare the watermark performance of WAPITI with traditional watermark distillation (WD) on base models. Table 2 presents the overall results. In this experiment, WAPITI uses watermark parameters extracted from a model trained with only 1000 steps of watermark distillation, while WD models are trained for 5000 steps. The watermark coefficient $\lambda_{FT}$ is set to 1.5, which provides a balance between watermark detectability and generation quality.

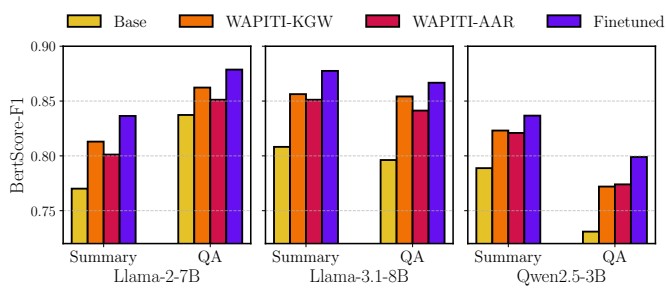 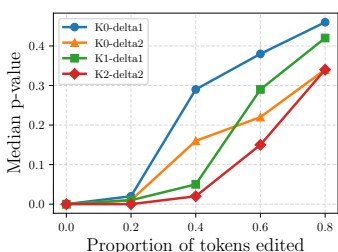

Figure 3: WAPITI compared with base and fine-tuned models. WAPITI preserves the capabilities of the fine-tuned model, achieving performance close to the original finetuned models.

Figure 4: Watermark p-values of generations with various proportions of random corruption.

The results show that WAPITI achieves watermark detectability comparable to, and in some cases better than, full watermark distillation, while largely preserving model generation quality. For the KGW watermark, WAPITI even surpasses WD in detectability, though with slightly higher perplexity. However, the seq-rep-3 values remain nearly identical, indicating that the watermark parameters avoid the common repetition issues seen in watermarked models. But for the AAR watermark, WAPITI shows slightly inferior TPR@0.05, which is still significantly distinctive.

In terms of training cost, WAPITI is significantly more efficient. Because watermark strength can be enhanced by adjusting the watermark coefficient $\lambda_{FT}$, fewer distillation steps are needed to extract meaningful watermark parameters. Our experiments show that 1000 steps are sufficient, reducing training cost by about 80% compared with traditional WD. Moreover, the extracted watermark parameters can be reused to transfer watermarks to other fine-tuned models, further reducing the per-model cost. For example, when applied across four different models, the average cost of watermarking per model is only 5% of that required by traditional distillation.

| Model | Task | $p$-value ($\downarrow$) | | TPR@0.1 ($\uparrow$) | | TPR@0.05 ($\uparrow$) | | Perplexity / seq-rep-3 ($\downarrow$) | |
|---|---|---|---|---|---|---|---|---|---|
| | | KGW | AAR | KGW | AAR | KGW | AAR | KGW | AAR |
| Llama-2 | QA | $2.32 \times 10^{-15}$ | $1.57 \times 10^{-12}$ | 1.00 | 0.91 | 1.00 | 0.89 | 8.13/0.03 | 22.41/0.09 |
| | Instruct | $3.39 \times 10^{-09}$ | $3.92 \times 10^{-13}$ | 1.00 | 0.90 | 1.00 | 0.85 | 4.96/0.06 | 15.70/0.05 |
| | Sum | $4.67 \times 10^{-13}$ | $6.84 \times 10^{-13}$ | 1.00 | 0.90 | 1.00 | 0.86 | 8.11/0.09 | 19.54/0.06 |
| Llama-3 | QA | $4.00 \times 10^{-21}$ | $1.20 \times 10^{-14}$ | 1.00 | 0.87 | 1.00 | 0.84 | 14.89/0.03 | 18.51/0.04 |
| | Instruct | $5.20 \times 10^{-17}$ | $3.27 \times 10^{-13}$ | 1.00 | 0.88 | 1.00 | 0.86 | 17.94/0.03 | 18.53/0.04 |
| | Sum | $1.18 \times 10^{-06}$ | $7.80 \times 10^{-15}$ | 1.00 | 0.88 | 1.00 | 0.86 | 14.05/0.06 | 16.02/0.06 |
| Qwen-2.5 | QA | $9.36 \times 10^{-05}$ | $8.52 \times 10^{-14}$ | 1.00 | 0.92 | 1.00 | 0.90 | 18.09/0.01 | 19.28/0.03 |
| | Instruct | $8.50 \times 10^{-07}$ | $2.41 \times 10^{-13}$ | 1.00 | 0.90 | 1.00 | 0.87 | 16.78/0.02 | 18.27/0.04 |
| | Sum | $3.21 \times 10^{-05}$ | $1.65 \times 10^{-13}$ | 1.00 | 0.91 | 1.00 | 0.89 | 21.14/0.02 | 20.50/0.03 |

Table 3: Results of WAPITI on fine-tuned models with KGW and AAR. Each task corresponds to the fine-tuning type. TPR@0.1 denotes the true positive rate at a false positive rate of 0.1.

### Watermark Detectability

Table 3 reports the results for watermark detectability and generation quality on three different finetuned models to prove its compatibility with finetuned models. For the main comparison, we focus on the KGW ($k = 1, \gamma = 0.25$) and AAR ($k = 2$) watermark settings; full experimental results are provided in Appendix E. For WAPITI, all watermark parameters are extracted from a model trained with 1000 steps of watermark distillation, and we select the most suitable watermark coefficient $\lambda_{FT}$ to balance detectability and generation quality.

The results show that both watermark types achieve strong detectability after transfer. The near-perfect TPR scores (close to 1.0) indicate that the watermarks are highly reliable. In addition, model

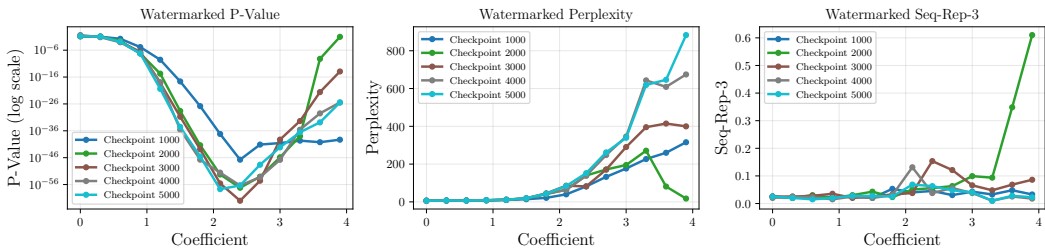

Figure 5: Effect of watermark coefficient $\lambda_{FT}$ on detectability and generation quality.

generation quality is largely preserved: perplexity shows only minor increases, while seq-rep-3 remains almost unchanged.

When comparing across models, we observe that different architectures respond differently to watermark parameters. Between the two watermark schemes, KGW consistently outperforms AAR in both detectability and generation quality. We attribute this to the relative complexity of the watermarking process: KGW modifies logits directly, which can be interpreted as learning a "vocabulary preference" during distillation. In contrast, AAR involves more complex semantic encoding, making it harder for the model to fully capture during distillation and subsequent parameter transfer.

**Fine-tuned Capability.** Figure 3 shows model performance before and after fine-tuning. The WAPITI fine-tuned models achieve gains in both question answering and summarization tasks compared with the base model, and their performance remains close to that of models fine-tuned without watermarking. This demonstrates that WAPITI is compatible with the fine-tuning process.

To ensure fairness, we distilled watermarks using the RealNewsLike subset of the C4 dataset, applying careful deduplication with LLM assistance. For evaluation, we specifically selected Medical QA and legal summarization tasks, which introduce domain-specific knowledge not contained in the base models. This ensures that performance improvements in the WAPITI fine-tuned models are derived from the fine-tuning process itself, rather than from watermark parameters.

# 6 ANALYSIS

In this section, we further analyze the mechanism behind WAPITI. We focus on three aspects: (i) the role of the watermark coefficient $\lambda_{FT}$ and its effect on watermark detectability and generation quality, (ii) robustness of WAPITI against random-edit attacks, and (iii) why WAPITI remains compatible with fine-tuned models from a parameter-level perspective.

## 6.1 EFFECT OF THE WATERMARK COEFFICIENT

We first examine the role of the watermark integration coefficient $\lambda_{FT}$. By varying $\lambda_{FT}$, we measure how watermark detectability and generation quality change. For this experiment, we use the KGW watermark ($k = 1, \delta = 2$) and evaluate on three models: Llama-3.1-8B, Llama-2-7B, and Qwen2.5-3B. Complete results are provided in Appendix E.

Figure 5 shows the effect of increasing $\lambda_{FT}$. For watermark detectability, when $\lambda_{FT}$ is in the range $[0, 2]$, the p-value decreases along an exponential trend with respect to log probability, confirming that $\lambda_{FT}$ effectively controls watermark strength. For generation quality, when $\lambda_{FT}$ is small (approximately $\leq 0.2$), both perplexity and seq-rep-3 remain stable, indicating that model quality is preserved.

However, as $\lambda_{FT}$ increases further, both detectability and generation quality collapse, showing that watermark parameters begin to interfere strongly with the fine-tuned model. Therefore, $\lambda_{FT}$ provides a controllable trade-off between watermark detectability and generation quality in WAPITI.

|         | K0-delta1 | K0-delta2 | K1-delta1 | K1-delta2 | AAR-K2 | AAR-K3 | AAR-K4 |
|---------|-----------|-----------|-----------|-----------|--------|--------|--------|
| QA      | 0.1%      | 0.1%      | 0.1%      | 0.1%      | 0.1%   | 0.1%   | 0.1%   |
| Summary | 0.3%      | 0.5%      | 0.2%      | 0.2%      | 0.3%   | 0.3%   | 0.3%   |
| Instruct| 0.2%      | 0.2%      | 0.1%      | 0.2%      | 0.6%   | 0.5%   | 0.5%   |

Table 4: Cosine similarity between task vectors and watermark parameters, showing that watermark parameters are nearly orthogonal to task vectors.

## 6.2 ROBUSTNESS TO RANDOM EDITS

To test robustness, we apply random-edit attacks by replacing tokens in generated text with random alternatives and then evaluating watermark detectability. Details of the setup are given in Appendix D. As shown in Figure 4, watermark p-values remain statistically significant until the edit proportion reaches about 20%–30%. This indicates that WAPITI maintains strong robustness even under substantial perturbations.

## 6.3 PARAMETER-LEVEL ANALYSIS

To understand why WAPITI remains compatible with fine-tuned models, we analyze parameter directions following (Ilharco et al., 2023). In short, we compare watermark parameters with task vectors (parameter differences induced by fine-tuning) and find them to be nearly orthogonal (Table 4). This orthogonality explains why, at small $\lambda_{FT}$, the model retains both generation quality and fine-tuned capabilities.

## 7 RELATED WORK

**Text watermarking.** Earlier works in text watermarking typically embedded information through post-processing of texts, closely resembling steganography (Venugopal et al., 2011; Yang et al., 2021). More recent studies have shifted towards decoding-based watermarking, hiding information by perturbing the text during the decoding phase (Kirchenbauer et al., 2023; Aaronson, 2023; Zhu et al., 2024; Krishna et al., 2023; Kuditipudi et al., 2024; Zhao et al., 2023a; Christ et al., 2023; Wu et al., 2024; Liu & Bu, 2024; Giboulot & Teddy, 2024; Lu et al., 2024; Ren et al., 2024; Wang et al., 2024).

**Model interventions.** Beyond fine-tuning, researchers have explored parameter-level interventions to modify model behaviors. Key approaches include model patching (Goel et al., 2020; Ilharco et al., 2023; Murty et al., 2022; Sung et al., 2021), parameter editing (Mitchell et al., 2022a;b; Santurkar et al., 2021), and model alignment (Askell et al., 2021; Glaese et al., 2022; Kasirzadeh & Gabriel, 2022). Compared to retraining or fine-tuning, model intervention offers a more efficient way to introduce new capabilities into models.

## 8 CONCLUSION

In this paper, we tackle the challenge of watermarking fine-tuned open-source language models, where existing distillation methods are costly and incompatible with fine-tuning. We propose WAPITI, a training-efficient approach, thereby enabling reliable watermarking without the need for repeated distillation.

Our theoretical analysis and empirical evaluations on Llama-2-7B, Llama-3-8B, and Qwen-2.5-3B show that WAPITI preserves the fine-tuned performance of models on tasks such as medical QA and legal summarization while achieving strong detectability. These results demonstrate that parameter integration offers a practical and scalable path for watermarking open-source models.

Future work could further improve WAPITI by exploring watermarking strategies specifically designed for transfer, refining the extraction and injection of watermark parameters to minimize interference, and optimizing the distillation of the base watermark model to reduce resource demands. Together, these directions may enhance the robustness, efficiency, and applicability of watermarking in open-source ecosystems.

ETHICS STATEMENT.

The paper uses only publicly available datasets and evaluates in a transparent, responsible manner in accordance with the code of ethics of ICLR.

REPRODUCIBILITY STATEMENT.

To ensure reproducibility, for **datasets details**, we include detailed statistics and descriptions of the datasets in Sec. 5.2. For **experimental setup**, we include a detailed description of adopted evaluation metrics, machines, dataset splits, and hyperparameter settings in Section 5.1 and Appendix E.

LLM USAGE DISCLOSURE

We use GPT-4 to assist with grammar polishing and drafting some background text. All scientific claims, analyses, proofs, and experiments were verified and written by the authors. No experimental design, result interpretation, or mathematical content was generated by an LLM without author oversight.

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

## A  DETAILED DEFINITION OF WATERMARK

In this section, we provide formal definitions of the watermarking schemes used in this work: KGW (Kirchenbauer et al., 2023) and AAR (Aaronson, 2023).

**KGW.**  For the KGW watermark, we adopt the same notation as in the main text. Let $\mathcal{W}^{KGW}$ denote the watermarking algorithm, $f_{\boldsymbol{\theta}}(\cdot \mid \boldsymbol{x})$ the next-token distribution, and $\phi$ the watermark key. The hyperparameters $k, \gamma, \delta$ control the watermarking process: - $k$: number of preceding tokens used to construct the green list, - $\gamma$: proportion of the vocabulary assigned to the green list, - $\delta$: logit shift applied to green-list tokens.

The watermarked distribution is:

$$f_{\boldsymbol{\theta}}^{KGW}(\boldsymbol{x}, \phi, k, \gamma, \delta) = \text{softmax}\left(\log f_{\boldsymbol{\theta}}(\cdot \mid x) + \delta \cdot \mathcal{W}^{KGW}(x_{i-k}, \ldots, x_{i-1}; \phi; \gamma; |\mathcal{V}|)\right), \quad (9)$$

where $\mathcal{W}^{KGW}$ outputs a mask over the vocabulary indicating membership in the green list.

Detection is performed by counting green-list tokens and computing the corresponding binomial p-value:

$$\mathcal{D}^{KGW}(\boldsymbol{x}, \phi, \gamma) = 1 - \text{BinoCDF}\left(\sum_{t=1}^{|\boldsymbol{x}|} \mathcal{W}^{KGW}(x_{t-k}, \ldots, x_{t-1}; \phi; \gamma; |\mathcal{V}|)\right), \quad (10)$$

where BinoCDF is the cumulative distribution function of the binomial distribution.

**AAR.**  For the AAR watermark, we again let $\mathcal{W}^{AAR}$ denote the algorithm, $f_{\boldsymbol{\theta}}(\cdot \mid \boldsymbol{x})$ the next-token distribution, and $\phi$ the watermark key. AAR has a single hyperparameter $k$, the number of preceding tokens used to generate pseudorandom scores:

$$\boldsymbol{r}_i = \mathcal{W}^{AAR}(x_{i-k}, \ldots, x_{i-1}, \phi) \sim \text{Unif}(0,1)^{|\mathcal{V}|}. \quad (11)$$

Token generation follows the Gumbel–Max rule:

$$x_i^{AAR} = \arg\max_{j \in \mathcal{V}} \left(\log f_{\boldsymbol{\theta}}(j \mid x) - \log(-\log r_i^j)\right). \quad (12)$$

Detection is based on the cumulative score statistics, evaluated under a gamma distribution:

$$\mathcal{D}^{AAR}(\boldsymbol{x}, \phi) = 1 - \text{GammaCDF}\left(\sum_{t=k+1}^{|\boldsymbol{x}|} -\log\left(1 - \mathcal{W}^{AAR}(x_{t-k}, \ldots, x_{t-1}, \phi)_{x_t}\right); |\boldsymbol{x}| - k, 1\right), \quad (13)$$

where GammaCDF is the cumulative distribution function of the Gamma distribution, and the underbraced term corresponds to the score assigned to token $x_t$.

## B  METHOD DETAILED DERIVATION

First we observe that the parameter difference between the fine-tuned model and the base model, $\boldsymbol{\theta}_{\text{FT}} - \boldsymbol{\theta}_{\text{Base}}$, is approximately orthogonal to the parameter difference caused by watermarking, $\boldsymbol{\theta}_{\text{Base}}^{\dagger} - \boldsymbol{\theta}_{\text{Base}}$:

$$\langle \boldsymbol{\theta}_{\text{FT}} - \boldsymbol{\theta}_{\text{Base}}, \boldsymbol{\theta}_{\text{Base}}^{\dagger} - \boldsymbol{\theta}_{\text{Base}} \rangle \approx 0. \quad (14)$$

Let $\otimes$ denote the tensor product between differentiation operators, and let $\times_1, \times_2$ denote the mode-1 and mode-2 tensor–matrix product, respectively. Let $\boldsymbol{H}_{\text{Base}}(\boldsymbol{x}) := \nabla_{\boldsymbol{\theta}} \otimes \nabla_{\boldsymbol{\theta}} f_{\boldsymbol{\theta}_{\text{Base}}}(\boldsymbol{x})$ be the Hessian. As shown in prior studies, every channel of $\boldsymbol{H}_{\text{Base}}(\boldsymbol{x})$ is approximately the identity matrix $\boldsymbol{I}$ (Jiao et al., 2024; Yang et al., 2024). Combining it with our observation in Eq. (14), we hypothesize that:

$$\boldsymbol{H}_{\text{Base}}(\boldsymbol{x}) \times_1 (\boldsymbol{\theta}_{\text{FT}} - \boldsymbol{\theta}_{\text{Base}}) \times_2 (\boldsymbol{\theta}_{\text{Base}}^{\dagger} - \boldsymbol{\theta}_{\text{Base}}) \approx 0. \quad (15)$$

The first-order Taylor expansion of $\nabla_{\boldsymbol{\theta}} f_{\boldsymbol{\theta}_{\text{FT}}}(\boldsymbol{x})$ around $\boldsymbol{\theta} = \boldsymbol{\theta}_{\text{Base}}$ is:

$$\nabla_{\boldsymbol{\theta}} f_{\boldsymbol{\theta}_{\text{FT}}}(\boldsymbol{x}) = \nabla_{\boldsymbol{\theta}} f_{\boldsymbol{\theta}_{\text{Base}}}(\boldsymbol{x}) + \boldsymbol{H}_{\text{Base}}(\boldsymbol{x}) \times_1 (\boldsymbol{\theta}_{\text{FT}} - \boldsymbol{\theta}_{\text{Base}}) + O(\|\boldsymbol{\theta}_{\text{FT}} - \boldsymbol{\theta}_{\text{Base}}\|^2), \quad (16)$$

$$\boldsymbol{H}_{\text{Base}}(\boldsymbol{x}) \times_1 (\boldsymbol{\theta}_{\text{FT}} - \boldsymbol{\theta}_{\text{Base}}) \approx \nabla_{\boldsymbol{\theta}} f_{\boldsymbol{\theta}_{\text{FT}}}(\boldsymbol{x}) - \nabla_{\boldsymbol{\theta}} f_{\boldsymbol{\theta}_{\text{Base}}}(\boldsymbol{x}). \quad (17)$$

Next, substituting Eq. (17) into Eq. (15), we find that the gradient difference between the fine-tuned and base models, when multiplied by the watermarked parameter difference of the base model, is approximately zero:

$$(\nabla_{\boldsymbol{\theta}} f_{\boldsymbol{\theta}_{\text{FT}}}(\boldsymbol{x}) - \nabla_{\boldsymbol{\theta}} f_{\boldsymbol{\theta}_{\text{Base}}}(\boldsymbol{x}))\Delta\boldsymbol{\theta}_{\text{Base}} \approx 0. \quad (18)$$

By rearranging Eq. (18), we conclude that the gradients of the fine-tuned and base models are approximately equal when applied to the watermarked parameter difference:

$$\nabla_{\boldsymbol{\theta}} f_{\boldsymbol{\theta}_{\text{FT}}}(\boldsymbol{x})\Delta\boldsymbol{\theta}_{\text{Base}} \approx \nabla_{\boldsymbol{\theta}} f_{\boldsymbol{\theta}_{\text{Base}}}(\boldsymbol{x})\Delta\boldsymbol{\theta}_{\text{Base}}. \quad (19)$$

## C  LIMITATION OF WATERMARK DISTILLATION

**Dataset.**  We use the `multi-lexsum` dataset, which contains 9.2k legal case summaries. For watermark distillation, we use the `RealNewsLike` split from the C4 (Raffel et al., 2020) dataset.

**Fine-tuning.**  Fine-tuning is performed using the `Trainer` API from `Transformers`. We train for 500 steps with a batch size of 8, using `bf16` for mixed-precision training. Optimization is carried out with DeepSpeed ZeRO-2. Each run takes about 1 hour on 8 NVIDIA A100 80GB GPUs.

**Distillation.**  For watermark distillation, we train for 5,000 steps with a batch size of 16. The first 50 tokens of each sample are used as the prompt, and the maximum generation length is set to 200.

**Watermarked Dataset Generation.**  To construct the watermarked dataset, we use Llama-3.1-70B. Generation is performed with deterministic decoding (no sampling).

## D  ROBUSTNESS TO RANDOM TEXT EDIT

In this experiment, we use the WAPITI model to generate samples from OpenWebText prompts under the KGW watermark with $k \in \{0, 1\}$, $\gamma = 0.25$, and $\delta \in \{1, 2\}$. The generation length is fixed at 200 tokens. We then apply random edits with proportions $\epsilon \in \{0.1, 0.2, \ldots, 0.8\}$, where each token has probability $\epsilon$ of being replaced by a random token. After editing, we compute the median p-value for watermark detection. Results are shown in Fig 4

## E  WAPITI EXPERIMENT

### E.1  TRAINING PARAMETERS

We fine-tune models using AdamW with a peak learning rate of $2\text{e}^{-5}$, linear warmup over the first 10% of steps, and cosine decay scheduling. Training runs for 5,000 steps with a global batch size of 16. Prompts are constructed by taking the first 50 tokens of each sample, and `max_length` is set to 200.

Gradient clipping is applied at 1.0, weight decay is set to 0.1, and training is performed in bf16 mixed precision. For efficiency, we use FSDP optimization. Training Llama-2-7B and Llama-3.1-8B each takes about 4 hours on 8 NVIDIA A100 80GB GPUs, while Qwen2.5-3B requires about 3 hours on 2 NVIDIA A800 80GB GPUs.

### E.2  EXPERIMENT RESULTS

Following are the full experiment results.

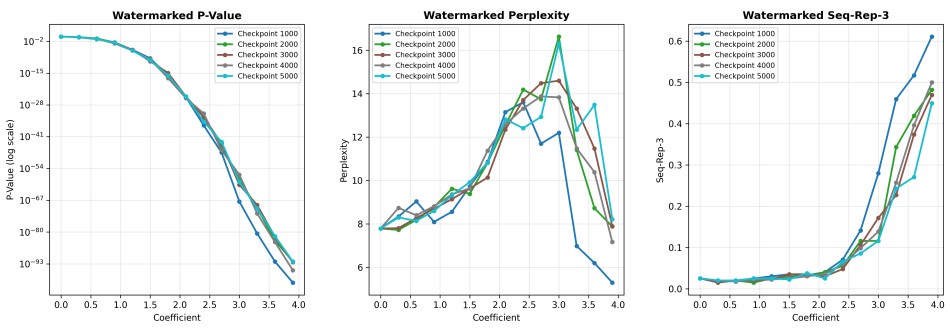

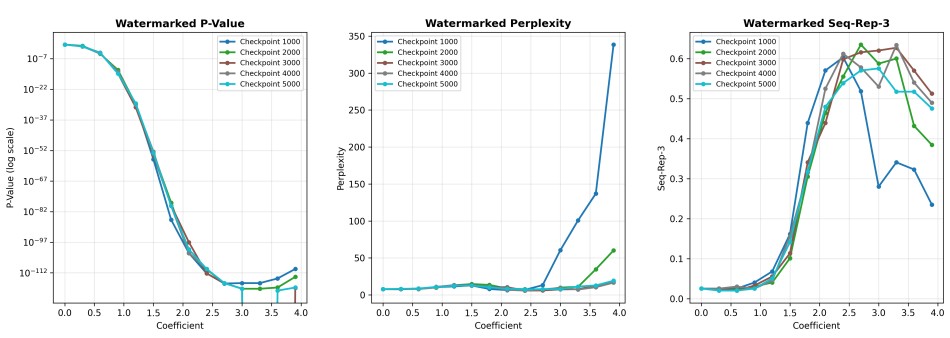

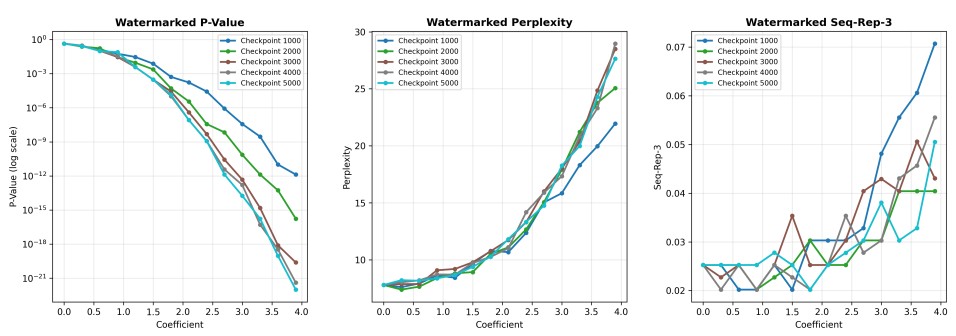

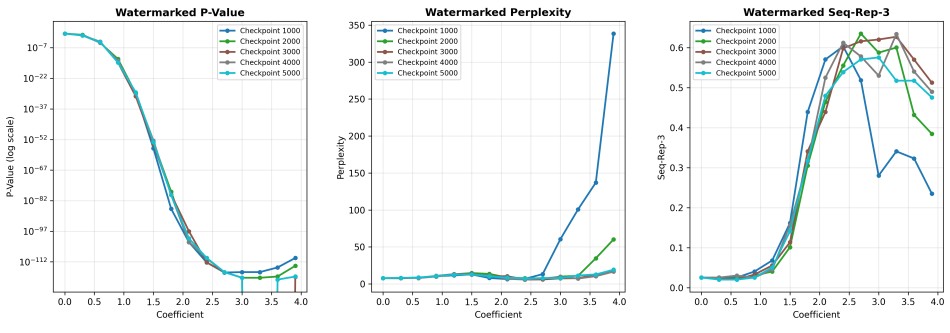

**Watermark Vector Analysis**
**Model: Llama_2_7b_hf**
**Watermark: Llama_2_7b_hf_logit_watermark_distill_kgw_k0_gamma0.25_delta1**

**Watermark Vector Analysis**
**Model: Llama_2_7b_hf**
**Watermark: Llama_2_7b_hf_logit_watermark_distill_kgw_k0_gamma0.25_delta2**

**Watermark Vector Analysis**
**Model: Llama_2_7b_hf**
**Watermark: Llama_2_7b_hf_logit_watermark_distill_kgw_k1_gamma0.25_delta1**

**Watermark Vector Analysis**
**Model: Llama_2_7b_hf**
**Watermark: Llama_2_7b_hf_logit_watermark_distill_kgw_k1_gamma0.25_delta2**

**Watermark Vector Analysis**
**Model: Meta_Llama_3.1_8B**
**Watermark: Meta_Llama_3.1_8B_logit_watermark_distill_kgw_k0_gamma0.25_delta1**

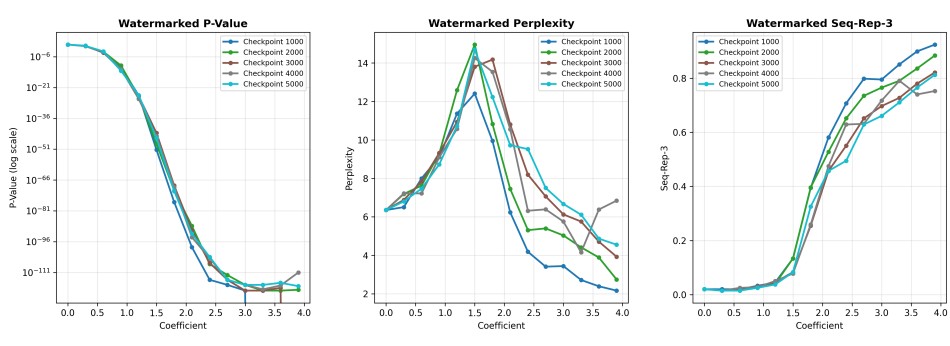

**Watermark Vector Analysis**
**Model: Meta_Llama_3.1_8B**
**Watermark: Meta_Llama_3.1_8B_logit_watermark_distill_kgw_k0_gamma0.25_delta2**

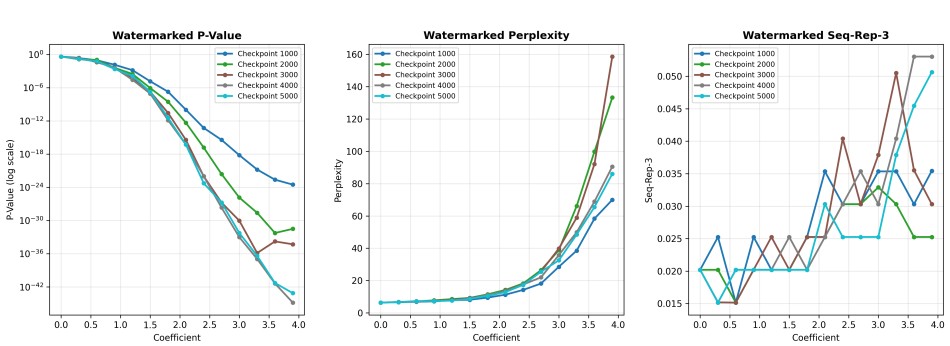

**Watermark Vector Analysis**
**Model: Meta_Llama_3.1_8B**
**Watermark: Meta_Llama_3.1_8B_logit_watermark_distill_kgw_k1_gamma0.25_delta1**

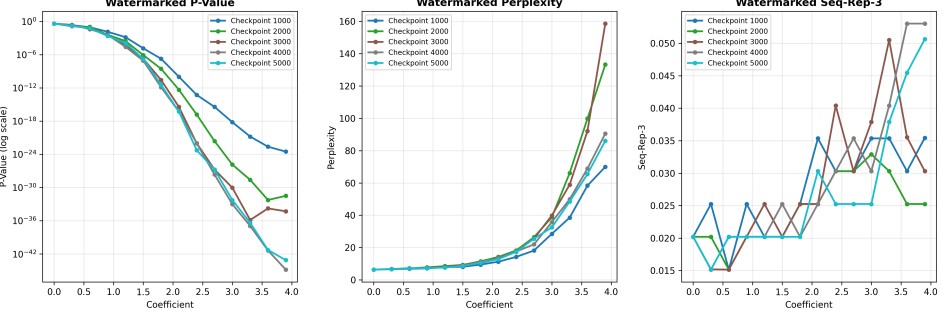

**Watermark Vector Analysis**
**Model: Meta_Llama_3.1_8B**
**Watermark: Meta_Llama_3.1_8B_logit_watermark_distill_kgw_k1_gamma0.25_delta2**

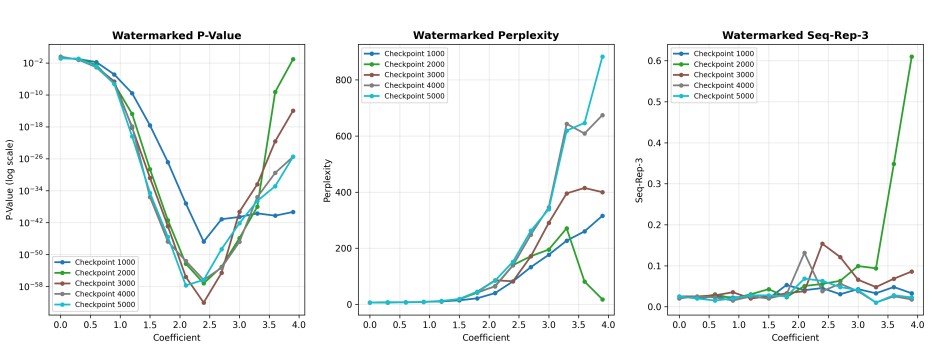

