# OpenReview forum: "WAPITI: A Watermark for Finetuned Open-Source LLMs"
_ICLR.cc/2026/Conference — Submitted to ICLR 2026_

### Official Review · Reviewer_HGvX · 2025-10-28

**Soundness:** 2
**Presentation:** 3
**Contribution:** 2
**Rating:** 4
**Confidence:** 4

**Summary:**

## Summary

This paper introduces WAPITI, a framework designed to embed detectable watermarks into fine-tuned open-source large language models. The key idea is to transfer a watermark signal from a watermarked base model to any fine-tuned model without retraining. The process involves first performing watermark distillation on the base model to obtain a parameter difference that captures the watermark effect. This parameter difference is then integrated into the parameters of a fine-tuned model using a scaling factor, producing a watermarked version of the fine-tuned model. This operation is performed directly in parameter space and does not require access to the fine-tuning data.

The paper provides a theoretical justification showing that this integration approximates the outcome of full watermark distillation while avoiding its computational cost. The results indicate that WAPITI preserves watermark detectability and model performance, achieving strong detection rates and maintaining comparable scores in BERTScore-F1 and perplexity. The method is also reported to be robust under text perturbations such as random word replacements.

**Strengths:**

## Strengths

1. The proposed approach can be applied through a single parameter integration step. It efficiently transfers the watermark from the base model to multiple fine-tuned derivatives, reducing computational cost.

2. The experiments show that the watermarked models maintain nearly the same performance as the original fine-tuned models while achieving strong watermark detectability.

3. The paper also provides a clearer mathematical explanation for why parameter integration preserves both watermark signal and model capability, supported by gradient-based derivations and empirical verification of parameter direction orthogonality.

**Weaknesses:**

## Weaknesses

1. Although parameter integration successfully transfers the watermark from the base model to fine-tuned models, the paper does not evaluate whether the watermark persists after further fine-tuning on non-watermarked data. The current robustness evaluation only covers surface-level perturbations such as random text edits, but not parameter-space changes caused by additional fine-tuning. Moreover, there is no mechanism proposed to preserve the watermark during subsequent training, such as gradient projection, regularization, or structured freezing. In essence, the method demonstrates “transferability” but not “durability.”

2. In open-source settings, an attacker may have access to both the unwatermarked base model and the watermarked fine-tuned model, allowing them to approximate watermark removal by computing or subtracting the weight difference. Although the paper claims that removing the watermark should significantly degrade model capability, it does not design or simulate such watermark-removal attacks, nor does it provide a “detectability–capability” trade-off curve. Furthermore, it lacks discussion of realistic release strategies (e.g., releasing only watermarked weights, withholding the base model) and whether these strategies are feasible or sufficient in practice.

3. The baseline coverage remains narrow. The experiments primarily focus on KGW and AAR watermarking schemes, without including more representative or challenging baselines such as KTH, which is designed for low-entropy text. This limitation makes it difficult to generalize the conclusions or to compare WAPITI comprehensively against state-of-the-art watermarking methods.

**Questions:**

The paper builds its methodological validity on two key assumptions: the near-orthogonality between the fine-tuning direction and the watermark direction, and the adequacy of a first-order approximation. However, it does not specify the conditions under which these assumptions might fail or provide counterexamples. For instance, if the downstream fine-tuning direction becomes increasingly correlated with the watermark direction, would the watermark be more vulnerable to erosion or partial loss?

---

### Official Review · Reviewer_JVc6 · 2025-10-30

**Soundness:** 2
**Presentation:** 2
**Contribution:** 2
**Rating:** 2
**Confidence:** 4

**Summary:**

This paper focuses on how to watermark the text of open-source LLMs. Prior work (Gu et al., 2024) finds that "baking" in watermarks through finetuning can be undone by further finetuning. WAPITI proposes to "bake" watermarks into open-source LLMs by merging (?) it with its finetuned, watermarked version. Results show that merging watermarked weights allows the model to retain the watermark, and follow the argumentation (model retains performance + retains watermark).

**Strengths:**

S1. Results seem fine. It’s cool that you can merge the model weights and the watermark still persists.

**Weaknesses:**

W1. I am not convinced of the setting. From an adversarial perspective, why would someone maliciously finetuning Llama later add in the watermarked version of the model? Doesn’t make sense to me and is probably why Gu et al., 2024 never considered this setting. I can only see a partial argument of saving costs for watermark tuning for other people down the line.

W2. Why does the assumption in eq. (1) hold? Is the intuition here due to the law of large numbers? (i.e. dot products are sums of many random numbers?) I looked in the appendix and skimmed Ilharco 2023 and can’t find why this is valid. Even if you accept this assumption, this seems extremely strong to me and would “solve” all of model merging through plain addition. Something is wrong here.

**Questions:**

What is \delta theta_base? I think its the difference of the base model with the watermarked base model but I don’t see it defined anywhere. I think this paper is not ready for presentation, and still needs to be worked on. I do not recommend acceptance at this stage.

---

### Official Review · Reviewer_9geF · 2025-10-30

**Soundness:** 3
**Presentation:** 3
**Contribution:** 2
**Rating:** 4
**Confidence:** 3

**Summary:**

This paper addresses the incompatibility of existing watermark distillation techniques with fine-tuned open-source LLMs. Current methods either lose the watermark upon fine-tuning or cause the model to forget its specialized abilities, and repeatedly applying distillation is computationally expensive. The authors propose WAPITI, a training-efficient framework. The approach involves performing watermark distillation once on a base model to extract "watermark parameters," which represent the parametric shift caused by watermarking. Experiments on Llama and Qwen models show WAPITI achieves strong detectability, with a TPR up to 1.00 at a 0.05 FPR, while preserving performance on downstream tasks like medical question answering and legal summarization.

**Strengths:**

1.  The method’s main contribution is transferring watermarks via parameter arithmetic. This leverages a single base model distillation to efficiently watermark multiple fine-tuned variants without individual retraining (Algorithm 1).
2.  Experimental validation is extensive. It covers multiple model families (Llama, Qwen), diverse fine-tuning tasks, and two distinct watermarking algorithms (KGW, AAR), demonstrating broad applicability (Tables 2, 3).

**Weaknesses:**

1.  The core method relies on a key theoretical derivation (Section 4 and Appendix B), but the foundational assumption, particularly in Equation (15), lacks sufficient mathematical justification. The authors should either clarify the origin of this assumption more rigorously or frame the derivation as a heuristic theoretical model rather than a strict proof.
2.  The derivation repeatedly uses first-order Taylor expansions for approximation, but fails to discuss the conditions under which these approximations are valid or analyze the potential errors introduced by higher-order terms. A discussion on the norm of \(\Delta \theta_{\mathrm{Base}}\) and the validity of the approximations is recommended.
3.  The paper claims "broad compatibility" with various watermarking schemes, yet experimental results show that its effectiveness with KGW is significantly better than with AAR. The authors are advised to conduct a deeper analysis of the root causes for this performance discrepancy; for instance, is it because KGW's logit-shift is inherently more "linear" and thus more amenable to transfer via parameter addition?

**Questions:**

1.  Could you provide more intuition or theoretical support for the assumption in Equation (15)? Specifically, why does the approximate orthogonality of the task vector and the watermark vector imply that their product with the Hessian tensor is approximately zero?
2.  Your experimental results indicate that WAPITI is more effective at transferring the KGW watermark than the AAR watermark. Could you elaborate on the potential reasons for this difference? Does the transferability of a watermark depend on how it manipulates the output distribution (e.g., direct logit shifts vs. complex modifications to the sampling process)?
3.  In the experiments on fine-tuned models (Table 3), you mention selecting "the most suitable watermark coefficient \(\lambda_{FT}\)". What was the specific procedure for determining this "most suitable" value? Was a validation set used to tune this hyperparameter for each model/task pair? This is crucial for understanding the practical overhead of the method.

---

### Official Review · Reviewer_pcos · 2025-10-31

**Soundness:** 2
**Presentation:** 3
**Contribution:** 2
**Rating:** 2
**Confidence:** 5

**Summary:**

This paper proposes a method called WAPITI for watermark reuse in fine-tuning scenarios of open-source large language models. Since traditional distillation-based watermarking methods are prone to failure after fine-tuning, and re-distilling the watermark for each fine-tuned model is costly, WAPITI first performs watermark distillation on the base model to obtain the parameter difference vector $\Delta\theta$ between the watermarked model and the original model. Then, $\Delta\theta$ is linearly injected into any fine-tuned model with an adjustable coefficient $\lambda$, thus quickly obtaining the watermarked version without retraining.

**Strengths:**

The paper proposes a parameter reuse approach fir watermarking fine-tuned large language models that significantly reduces the cost of watermark deployment.

**Weaknesses:**

1. The technical innovation is limited. The core idea of ​​WAPITI is to perform a watermark distillation on the base model to obtain the parameter difference vector $\Delta\theta$ between the watermarked model and the original model, and then linearly inject this vector into an arbitrarily fine-tuned model ($\theta_{FT}$ + $\lambda\cdot\Delta\theta$) with an adjustable coefficient $\lambda$. This parameter direction transfer approach is essentially similar to existing task vector, model merging, or adapter fusion methods, only replacing the task vector with the watermark vector. The paper does not propose new optimization objectives, algorithmic mechanisms, or detection principles, but rather applies existing parameter fusion ideas to the watermarking scenario. Overall, it is an engineering extension rather than an algorithmic innovation.

2. The authors explain why linear injection does not significantly affect task performance based on the premise that "the watermark vector and the task vector are approximately orthogonal." However, this conclusion is only based on empirical observations of a limited number of tasks (MedicalQA, LexSum). The paper lacks theoretical analysis explaining how $\Delta\theta$ injection affects model distribution or performance, and does not verify whether this orthogonality still holds under different tasks or model architectures. Furthermore, the method implicitly assumes a crucial premise: the corpus distribution of the fine-tuning task and the base model should maintain a certain similarity; otherwise, $\Delta\theta$ may conflict with the task update direction, thereby weakening task performance or even destroying the watermark signal. The paper does not discuss this similarity requirement, nor does it measure the relationship between task distribution differences and performance degradation. In other words, the effectiveness of WAPITI depends on the similarity of the distributions between the fine-tuning task and the base model, and this assumption needs explicit explanation and verification.

3. The paper claims that the method performs well in terms of robustness, but the experiments are limited to random token substitution (10–30%) and $\lambda$ coefficient scanning. These mild perturbations only demonstrate the model's robustness to surface noise and cannot cover more realistic or threatening attack scenarios, such as paraphrasing, re-distillation, LoRA overwriting, or parameter averaging. The lack of results under these stronger attacks makes the paper's conclusions on robustness too limited to support its practicality claims.

4. The paper observes that distribution-changing watermarking (KGW) outperforms distribution-preserving watermarking (AAR) in both detection and generation quality. The authors simply state that "KGW is easier to learn," but do not further analyze the mechanism. In fact, this phenomenon is understandable: KGW explicitly alters the text distribution during sampling (through logit shift), resulting in a more linear and stable output shift pattern for the teacher model, making it easier for the student model to learn during distillation. In contrast, AAR does not alter the text distribution during sampling, and the watermark signal is hidden in more complex pseudo-random perturbations or keyed semantic relationships, making it more difficult for the student model to fit. Therefore, it is reasonable that KGW is easier to learn than AAR, but the real challenge lies in improving the learnability of AAR. The paper does not explore which training or modeling factors determine the difficulty of AAR distillation. Specifically, I understand the performance differences between the various models as a combination of different factors: 1) The gap between Fine-tuned and WAPITI-AAR primarily reflects the impact of the distillation mechanism itself on text quality, because AAR does not change the output distribution, and the error of $\Delta\theta$ stems entirely from the imperfections of distillation learning. 2) The gap between Fine-tuned and WAPITI-KGW combines the effects of distillation error and the distortion probability shift of KGW. This analytical perspective helps me understand the experimental phenomena, but the paper does not quantitatively verify this, nor does it analyze which parameters are most susceptible to the influence of $\Delta\theta$ injection during the fine-tuning process.

5. The paper emphasizes that WAPITI can significantly reduce training costs, but it does not provide specific quantitative comparisons, such as the difference in GPU hours, time, or energy consumption for the distillations, nor does it show the overall cost curves under different numbers of tasks. This lack of data weakens the empirical support for the cost savings conclusion and diminishes its engineering significance.

**Questions:**

Please check weaknesses.

---

### Meta-Review · Area_Chair_HfMq · 2026-01-07

**Summary:**

All reviewers had concerns with this paper and did not recommend acceptance. Main concerns with this paper include limited novelty, weak evaluation and weak theoretical justifications. Transferring watermarks via parameter arithmetic is seen as straightforward. The evaluation is insufficient in terms of the tested attacks (which do not include paraphrasing, re-distillation or overwriting), some unexplained performance gaps between using the method with KGW and AAR and more experiments that would have been needed to support some of the author's claims.  Moreover, the key assumption of orthogonality between watermark and fine-tuning vectors lack justifications and is only empirically validated on a few tasks. The authors did not provide a rebuttal to these concerns, thus I recommend the paper's rejection.

**Reviewer Concerns:**

Not applicable since the reviewers did not provide a rebuttal.

**Reviewer Scores:**

Not applicable since there was no discussion from the author's side.

---

### Decision · Program_Chairs · 2026-01-26

Reject